# The antagonistic mechanism of *Bacillus velezensis* ZW10 against rice blast disease: Evaluation of ZW10 as a potential biopesticide

**Zuo Chen**[1,2]☯, **Lu Zhao**[3]☯, **Yilun Dong**[1,2], **Wenqian Chen**[1], **Chunliu Li**[1,2], **Xiaoling Gao**[1,2], **Rongjun Chen**[1,2], **Lihua Li**[1,2]*, **Zhengjun Xu**[1,2]*

**1** Rice Research Institute, Sichuan Agricultural University, Chengdu, China, **2** Crop Ecophysiology and Cultivation Key Laboratory of Sichuan Province, Chengdu, China, **3** Department of Bioengineering, Microbiology Laboratory of Sichuan Water Conservancy Vocational College, Dujiangyan, China

☯ These authors contributed equally to this work.
* lilihua1976@tom.com (LL); mywildrice@aliyun.com (ZX)

**Data Availability Statement:** All relevant data are within the manuscript and its Supporting Information files.

## Abstract

Rice blast, caused by the fungus *Magnaporthe oryzae*, is one of the three major diseases affecting rice production and quality; it reduces rice grain yield by nearly 30%. In the early stage of this study, a strain of *Bacillus velezensis* with strong inhibition of *M. oryzae* was isolated and named ZW10. *In vitro* assays indicated prolonged germination time of conidia of *M. oryzae* treated with the antifungal substances of ZW10, 78% of the conidia could not form appressorium, and the conidial tubes expanded to form vacuolar structure and then shrank. The results of FDA-PI composite dyes showed that the antifungal substances of ZW10 inhibited the normal activity of *M. oryzae* hyphae that were rarely able to infect the epidermal cells of rice leaf sheath *in vivo* tests. In addition, rice treated with the antifungal substances of ZW10 showed a variety of defense responses, including activation of defense-related enzymes, increased expression of the salicylic acid pathway genes, and accumulation of hydrogen peroxide ($H_2O_2$), which might function directly or indirectly in resistance to pathogen attack. The field experiment with rice blast infection in different periods showed that the antifungal substances of ZW10 had the same control effect as carbendazim. The significant biological control activity of ZW10 and its capacity to stimulate host defenses suggest that this *B. velezensis* strain has the potential to be developed into a biopesticide for the biocontrol of rice blast.

## Introduction

Rice (*Oryza sativa* L.) is one of the most important food crops in the world, being the staple food of about half of the world population [1]. However, rice blast, a devastating fungal disease caused by *Magnaporthe oryzae*, causes large losses in rice production and quality worldwide. When the rice blast is prevalent, the rice yield is reduced by 10–30%, and even 40–50% when the disease is severe [2–4]. *M. oryzae*, similarly to many other plant pathogens, is infecting not only rice, but also other crops, such as barley, wheat, sorghum, corn, and millet [5]. Rice blast

**Funding:** Currently, this work was supported by the Sichuan Science and Technology Program under grant number 2020YJ0352 and grant number 2020YJ0411.

**Competing interests:** The authors have declared that no competing interests exist.

affects different tissues of rice at different growth stages, with leaf blast and neck blast being the most common and most harmful [6, 7].

Synthetic fungicides play an important role in the control of rice blast. However, the overuse of synthetic fungicides has led to increasingly serious environmental pollution and potential health problems in humans and livestock, as well as pathogens developing resistance to the fungicides [8]. Hence, identification and use of bio-control agents to manage rice blast disease is gaining importance, with a growing interest in (i) the discovery and development of new and improved fungicides based on natural products that are environmentally friendly as well as (ii) the introduction of alternative measures such as biological control agents to manage plant diseases [9]. So far, a large number of microorganisms have been reported to have an antagonistic effect on rice blast. Gao et al. [10] isolated *Streptomyces albidoflavus* OsiLf-2 from rice that was antagonistic to *M. oryzae*. The fermentation filtrate of *Bacillus laterosporus* BPM3 controlled rice blast [11]. *Bacillus cereus* REN3 and REN4 isolated from rice rhizosphere and roots promoted rice growth and inhibited some rice pathogenic fungi [12].

*B. velezensis*, a relatively new species of *Bacillus*, was discovered in 2005 [13]. In recent years, the research on *B. velezensis* mainly focused on promoting the growth of animals and plants, induction of systemic resistance, production of antimicrobial substances, antagonism against pathogens, and the underlying mechanisms [14, 15]. *B. velezensis* FZB42 not only inhibited *Rhizoctonia solani* directly, but also mediated the defense response in lettuce [16]. Difficidin was purified and identified from the secondary metabolites of *B. velezensis* AP193, which has the effect of controlling tomato bacterial spot disease [17–19]. Zhang et al. [20] investigated the antifungal effects of non-volatile lipopeptides and volatile organic compounds released by *B. velezensis* C16 on the *Alternaria solani*. The surfactant A, which was isolated from *B. velezensis* by Jin et al. [21], had a strong inhibitory effect on *Xanthomonas oryzae* pv. *oryzae*.

Up to now, there are only a few reports on the control of rice blast by *B. velezensis*. Previous studies characterized the physical and chemical properties of *B. velezensis* ZW10 and elucidated its antagonistic activity against *M. oryzae* [22]. In this paper, we aimed to explore the inhibitory mechanism of *B. velezensis* ZW10 and the host defense response against *M. oryzae*, and to evaluate the potential of ZW10 as a biological pesticide.

## Materials and methods

### Determination of the concentration of antifungal substances

*B. velezensis* ZW10 was inoculated to 10 L of Landy medium (LM: 20 g glucose, 5 g L-glutamate, 10 g peptone, 1 g $KH_2PO_4$, 0.5 g KCl, 0.15 mg $FeSO_4$, 5 mg $MnSO_4$, 0.16 mg $CuSO_4$ in 1 L distilled water) and incubated at 35°C, 180 rpm on a rotary shaker for 72 h. The culture filtrate was extracted successively with 30 L of N-hexane, dichloromethane, ethyl acetate, and N-butanol. Each organic solvent was used for extraction three times. Each organic extract was concentrated to 50 mL using a rotary vacuum evaporator under reduced pressure of 42 mbar at 45°C to detect the antifungal activity. The organic extract with the strongest antifungal activity was chosen for purification using silica gel column chromatography. The mobile phase was $CH_2Cl_2$/MeOH at ratios of 100:0, 80:20, 60:40, 40:60, 20:80, and 0:100. The organic extract absorbed on the column was eluted by four column volumes of the mobile phase, then concentrated to 10 mL. The disc containing 9-day-old *M. oryzae* was placed in the center of PDA (Potato Dextrose Agar) plate, and then 20 μL fractions of crude-extract of fermentation broth (CFB) were added to determine the fraction with an antagonistic effect [23]. The active components were diluted with distilled water, and the half-inhibitory concentration (IC50) of CFB was determined by the plate confrontation method.

## Germination test of *M. oryzae* conidia and formation of appressorium

The *GFP*-tagged *M. oryzae* isolate (Guy11-Egfp) was kindly provided by State Key Laboratory of Crop Gene Exploration and Utilization in Southwest China. The conidia were scraped from 9-day-old Guy11-Egfp grown on complete medium (CM: 50 mL 20x nitrate salts, 1 mL trace elements, 10 g D-glucose, 2 g peptone, 1 g yeast extract, 1 g casamino acids, 1 mL vitamin solution, 15 g agar in 1 L distilled water) using the CFB at IC50 and distilled water; the spore concentration was adjusted to approximately $1 \times 10^5$ conidia/mL [24]. Then, 50 μL of spore suspension was dropped on the hydrophobic slide and kept at room temperature. After 2, 8, 12, 24, and 48 h, the germination rate of conidia and the formation of appressorium were observed under a ZEISS fluorescence microscope, with 100 conidia randomly selected for observation. The experiment was repeated three times.

The equation to calculate the germination rate was:

Germination rate (%) = $(A_1 / A_2) \times 100\%$ where $A_1$ = the number of germinated conidia, and $A_2$ = the total number of conidia.

The appressorium formation rate was calculated as follows:

Formation rate of appressorium (%) = $(B_1 / B_2) \times 100\%$ where $B_1$ = the number of conidia that formed appressorium and $B_2$ = the total number of conidia [25].

## Determination of *M. oryzae* mycelial activity

About 1 cm$^2$ agar disk containing mycelium of *M. oryzae* was put into 100 mL of CM, and cultured at 28˚C and 180 rpm for 3 days. The CFB at IC50 was added, and the other group was supplied with the equal amount of sterile water as control. After shaking for 24 h, the mycelia were stripped off *M. oryzae* cake, stained with FDA-PI composite dye, and treated in darkness at room temperature for 10 minutes [26]. After that, the stained hyphae were washed twice with PBS, sliced and observed under a ZEISS microscope. The antagonistic effect of *B. velezensis* on the morphological structure of *M. oryzae* mycelia was observed by scanning electron microscopy (SEM).

## Infection of rice leaf sheaths by *M. oryzae*

After 4 weeks of cultivation in greenhouse, we stripped the sheaths of the second leaf of the seedlings of susceptible rice variety Lijiangxintuanheigu (LTH). The spore suspension was prepared as described above. Then, 200 μL of spore suspension was slowly injected into rice leaf sheath [27]. At 2, 12, 24, and 36 h post-inoculation (hpi), the capacity of conidia to infect leaf sheath was observed under a ZEISS fluorescence microscope.

## Defense-related gene expression

The LTH plants were grown in a growth chamber (18 hours of light at 28˚C and 6 hours of darkness at 22˚C) to the three leaf–stage. First, plants were sprayed with the CFB at IC50 or distilled water with 0.1% Tween 20. After 24 h, plants were inoculated by spraying the conidia suspension (concentration of $1 \times 10^5$ conidia / mL) [10]. Rice leaves were collected at 0, 24, 48, and 72 hpi. Total RNA was extracted by the Trizol method, and the quality and quantity of RNA were determined by a Thermofisher Nano DROP. Amplification of cDNA was carried out using a Primescript RT reagent kit (Takara) by following the manufacturer's instructions. The qRT-PCRs were carried out using a BIO-RAD connect and normalized using *OsActin* expression levels as the internal reference. The primer sequences of the relevant defense genes and the internal reference gene are shown in Table 1. Real-time PCR was performed with

**Table 1. The primer sequences of defense-related genes and internal reference gene.**

| Gene | Forward primer (5'-3') | Reverse primer (5'-3') |
| --- | --- | --- |
| OsPR1a | GCTACGTGTTTATGCATGTATGG | TCGGATTTATTCTCACCAGCA |
| OsPR5 | GGTACAACGTCGCCATGAGCT | TGGGCAGAAGACGACTCGGTAG |
| OsPR10a | AATGAGAGCCGCAGAAATGT | GGCACATAAACACAACCACAA |
| OsWRKY45 | GCAGCAATCGTCCGGGAATT | GCCTTTGGGTGCTTGGAGTTT |
| OsLYP6 | TGCCCAGGACCACATCAGT | CCAGGGAAGCCCGGAATAT |
| OsPAL1 | CGAGTTCAACGCCGACAC | CCGGTAGAGCGGATACGAC |
| OsPOD | GGCCTTGGCAAATACCGACC | TCGTGTGTGCTCCTGAGAGA |
| OsActin | GAGTATGATGAGTCGGGTCCAG | ACACCAACAATCCCAAACAGAG |

SYBR Premix Ex Taq[TM] (TransGen Biotech, Beijing, China). All reactions were performed in triplicate.

## $H_2O_2$ accumulation

The LTH rice plants were treated as described above. For $H_2O_2$ accumulation, LTH rice leaves were collected at 0, 24, 48, and 72 hpi and stained with diaminobenzidine (DAB, 1 mg/mL) at pH = 3.8 in the dark for 12 h. The dyed leaves were destained with 95% ethanol until transparent, followed by rinsing in distilled water. The accumulation of $H_2O_2$ in leaves was observed under a ZEISS stereomicroscope.

## Field trial with the antifungal substances of ZW10 against leaf blast

The control ability of ZW10 against rice blast was studied in Wenjiang District, Chengdu City, Sichuan Province, China (30°68′ N, 103°85′ E). The field was divided into four blocks, each with an area of 1 m². To avoid the influence of different treatments a thin film was used to separate each plot. Each plot was evenly seeded with 100 seeds of Jiangnanxiangnuo (JNX) rice, and then cultivated for 30 days. The plots were sprayed with 150 mL of water, Landy medium, carbendazim or the CFB at IC50. All the above solutions contained 0.1% Tween 20. After 1 day, each plot was sprayed with 150 mL of the *M. oryzae* conidia suspension (concentration of $1 \times 10^5$ conidia / mL). Seven days after inoculation, 50 rice plants were collected using the five-point sampling method for disease index evaluation. The experiment had three replicates per treatment and was arranged in a randomized complete block design. The disease index was evaluated according to the Standard Evaluation System (SES) for rice [28].

Disease index (%) = ∑ (number of diseased plants in all disease categories × the value of the relevant level) / (total number of investigated plants × highest disease category) × 100.

## Field trial with the antifungal substances of ZW10 against neck blast

Field trial design and early rice seedling cultivation were the same as in the leaf blast experiment. When JNX seedlings were in the heading stage, the plots were sprayed with 150 mL water, Landy medium, carbendazim or the CFB at IC50. All the above solutions contained 0.1% Tween 20. After 1 day, each plot was sprayed with 150 mL of the *M. oryzae* conidia suspension (concentration of $1 \times 10^5$ conidia / mL). Thirty days after inoculation, 50 rice panicles were collected using the five-point sampling method for disease index evaluation. The experiment had three replicates per treatment and was arranged in a randomized complete block design. The disease index was evaluated according to the Standard Evaluation System (SES) for rice [28], as specified above.

## Statistical analysis

Statistical analysis of the data was evaluated with SPSS 20.0 software. The significant differences among treatment were assessed by one-way analysis of variance (ANOVA), followed by Duncan's multiple range tests (DMRT) when one-way ANOVA revealed significant differences. The *P*-Values < 0.05(*) and *P*-value < 0.01 (**) were considered to indicate statistical significance. All data was expressed as mean standard deviation.

## Results

### Purifying the fermentation broth of ZW10 and determining its IC50

The fermentation broth was extracted by organic solvents ranging in polarity from weak to strong. The dichloromethane, ethyl acetate and n-butanol extracts had antifungal activity, with ethyl acetate extract being the most active (Fig 1A). The ethyl acetate extract was separated and purified using a silica gel gradient; the activity of 60% MeOH extraction was the strongest (Fig 1B). Therefore, the IC50 of the 60% methanol fraction was determined by gradient dilution. As shown in Fig 1C, the inhibitory activity of the 60% methanol extract was decreasing due to gradient dilution. Finally, the concentration of 1% CFB was IC50.

### Effects of the antifungal substances of ZW10 on *M. oryzae* conidia germination and appressorium formation

The *M. oryzae* conidia were treated with distilled water or 1% CFB. As shown in Fig 2A, in the control group, the conidia began to produce germ tubes after 2 h. After 8 h, the germ tube elongated and appressorium formed at the other end of the tube. After treatment with 1% CFB, conidia germinated and formed germ tube, but longer time was needed compared with the control group. After 24 h, the hyphae were deformed and the middle part expanded to

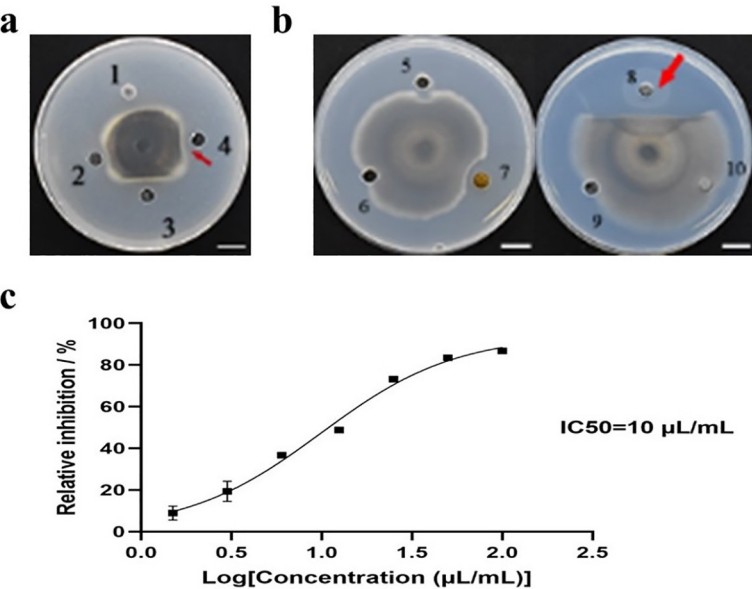

**Fig 1.** Isolation and IC50 of active components in the ZW10 cell-free culture filtrate extracted by different organic solvents (a, 1–4: N-hexane extract, dichloromethane extract, ethyl acetate extract, and N-butanol extraction), and the activity of various fractions of ethyl acetate extract in the $CH_2Cl_2$/MeOH system (b, 5–10 fractions were: 0% MeOH, 20% MeOH, 40% MeOH, 60% MeOH, 80% MeOH, and 100% MeOH). (c), IC50 of the 60% MeOH extraction against *M. oryzae*. Data are presented as means of three replicates ± SD. Scale bar, 10 mm.

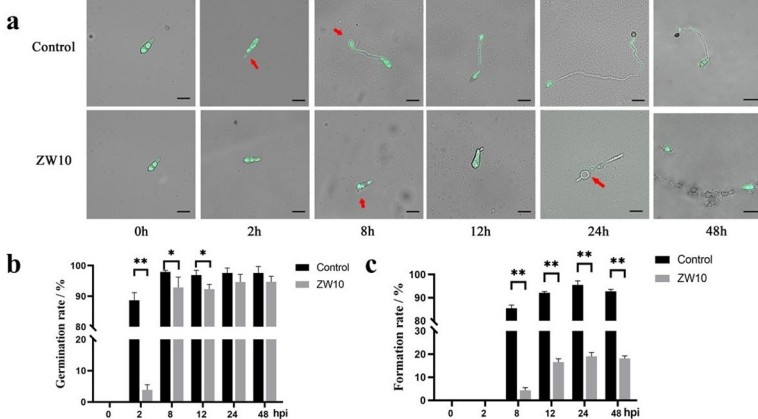

**Fig 2. Effects of the antifungal substances of ZW10 on *M. oryzae* conidia germination and appressorium formation.** Morphological changes in conidia of *M. oryzae* in different periods after 1% CFB of ZW10 treatment (a); germination rate (b) and appressorium formation rate (c) of *M. oryzae* conidia after ZW10 treatment. Data are presented as means of three replicates ± SD. **, the treatment difference was significant at P<0.01. Scale bar, 10 μm.

form vacuoles, followed by increasing the pressure in the vacuole. In the control group, the germination rate of conidia was 88.71 ± 2.01% at 2 h, and levelled off at 97.96 ± 0.42% after 8 h (Fig 2B). Appressorium began to form after 8 h, and the formation rate was 85.42 ± 1.15%, leavlling off at 92.07 ± 0.53% after 12 h. In the treatment group, the germination rate of conidia was only 3.87 ± 1.33% after 2 h, and higher than 90% after 8 h. However, only about 18% of conidia formed a normal appressorium structure (Fig 2C).

## Effects of the antifungal substances of ZW10 on *M. oryzae* membrane permeability and morphology

FDA dye can penetrate cell membrane and accumulate in living cells as green fluorescein, while dead cells are dyed red by PI [26]. The hyphae treated with distilled water showed obvious green fluorescence (Fig 3A). In contrast, the mycelia of *M. oryzae* treated with 1% CFB were stained with PI, and the red fluorescence was obvious, especially around the vacuoles.

The results of SEM (Fig 3B) showed that the hyphae treated with distilled water had smooth surface. After treatment with 1% CFB, mycelia of *M. oryzae* showed numerous cavitation structures, with some of them appearing broken; moreover, some hyphae were ruptured.

## Effect of the antifungal substances of ZW10 on *M. oryzae* conidial infection *in vivo*

In order to analyze the infection of rice by *M. oryzae*, the leaf sheath of LTH, a susceptible rice variety, was inoculated with *M. oryzae*, and fluorescence was assessed. During the formation of the specific appressoria structure of *M. oryzae*, glycerin accumulated in the inner part of the appressoria, resulting in huge swelling in the inner part until the formation of infection points, which allowed *M. oryzae* to colonize the rice epidermis [29–31]. As shown in Fig 4, at 12 hpi, the appressoria in control group differentiated to form melanin, and the appressoria's infection points pierced the rice epidermal cells. However, after treatment with 1% CFB, the spores formed germ tubes, but did not form appressoria. At 24 hpi, several secondary hyphae were derived from the inoculated hyphae in the control group and began to infect the adjacent epidermal cells, whereas in the experimental group, vacuoles were formed at the opposite end of

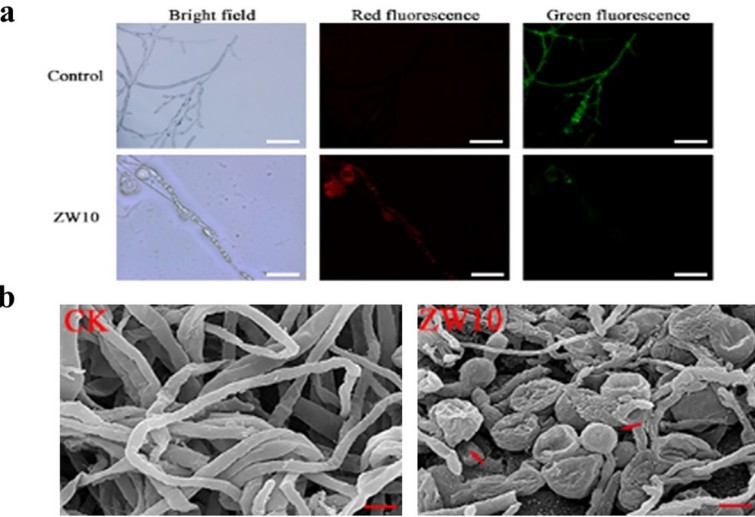

**Fig 3. The effect of the antifungal substances of ZW10 on *M. oryzae* membrane permeability and mycelium morphology.** After the treatment with 1% CFB of ZW10, the permeability of cell membrane was observed by FDA-PI staining (a), and the morphological structure of mycelium was observed by scanning electron microscopy (b). Scale bar in (a): 10 μm, and (b): 5 μm.

the conidia germ tube. At 36 hpi, the vacuoles in the experimental group began to collapse, and the deformed hyphae and shriveled vacuoles were observed in the leaf sheath epidermis.

## The antifungal substances of ZW10 enhanced rice resistance to *M. oryzae*

In most pathogen-plant interactions, systemic acquired resistance (SAR) becomes activated, which confers plants broad-spectrum resistance to persistent pathogen infection [32]. Thus, pathogenesis related proteins (PRs) genes (*OsPR1a*, *OsPR5*, *OsPR10a*), lysin motif-containing protein (LYP) gene (*OsLYP6*), phenylalanine ammonia-lyase (PAL) gene (*OsPAL1*), transcription regulation factor (TF) genes (*OsWRKY45*) and *OsPOD* gene were tested. *OsPR1*, *OsPR5*, *OsPR10a*, *OsPAL1*, *OsLYP6*, and *OsWRKY45* are known to be involved in the salicylic acid

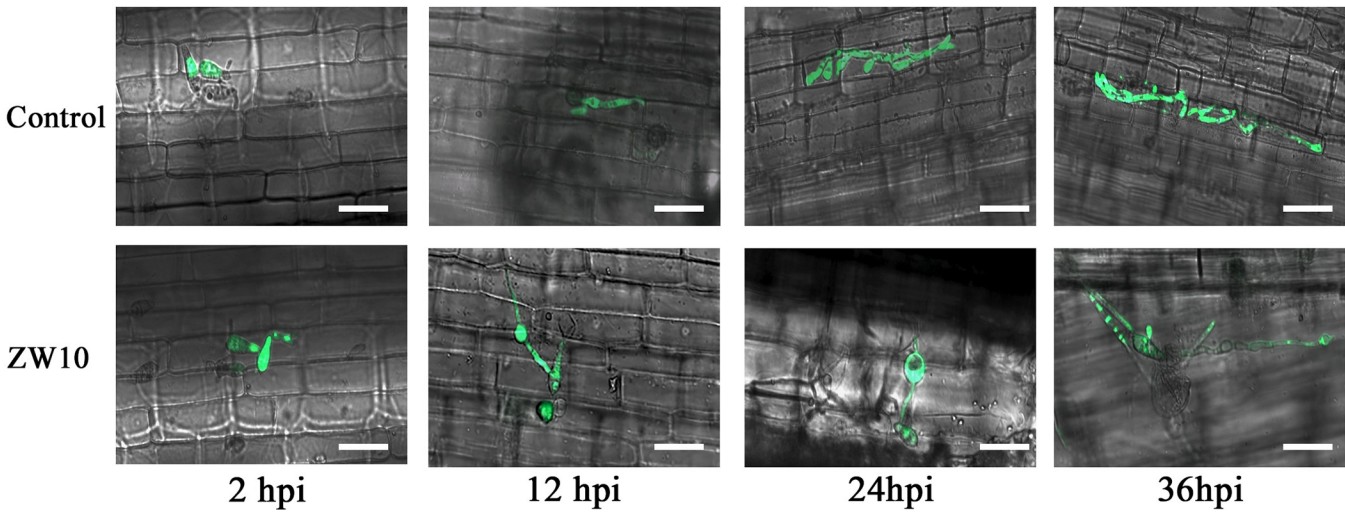

**Fig 4. The antifungal substances of ZW10 were antagonistic to *M. oryzae in vivo*.** After the treatment with 1% CFB of ZW10 treatment, the capacity of *M. oryzae* mycospores to infect epidermal cells of leaf sheath was altered. Scale bar, 20 μm.

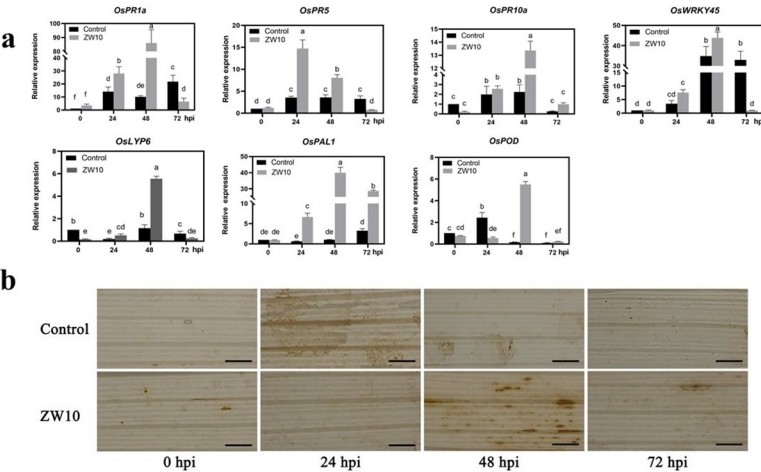

**Fig 5. The antifungal substances of ZW10 induced rice capacity to resist *M. oryzae*.** The expression of relevant defense genes in rice leaves after inoculation with 1% CFB of ZW10 (a), and differential content of $H_2O_2$ as indicated by diaminobenzidine (DAB) staining (b). Data are presented as means of three replicates ± SD. Means with different letters have significant differences (P < 0.05). Scale bar, 50 μm.

(SA) signaling pathway [33]. As shown in Fig 5A, the defense genes of rice were activated to varying degrees from 24 hpi to 72 hpi. The expression of defense genes in rice treated with 1% CFB was significantly increased. The peak expression of defense genes occurred at 24 hpi or 48 hpi. The highest expression of SA pathway-related genes was observed at 48 hpi, except for *OsPR5*, which reached the peak after 24 h. The Os*POD* gene reached maximum expression at 48 hpi.

H$_2$O$_2$ is a relatively stable ROS. When a host is infected by a pathogen, the accumulated H$_2$O$_2$ mediates the programmed cell death (PCD) of infected and surrounding cells [34]. The DAB staining method was used in this experiment, and the results are shown in Fig 5B. The rice leaves treated with fermentation broth gradually accumulated H$_2$O$_2$, reaching maximum at 48 hpi.

## Biocontrol efficacy of the antifungal substances of ZW10

Field experiments were conducted to further evaluate the control effect of *B. velezensis* on rice blast. The results are shown in Fig 6A. In the treatment with 1% CFB, the leaf blast disease index was 17.70 ± 1.89%, which was significantly lower than that of the water (80.30 ± 6.1%) and the Landy culture (79.18±5.8%) treatments. The control efficiency of 1% CFB was similar to that of the carbendazim treatment (17.72 ± 2.38%), a fungicide (Fig 6B).

Rice neck blast is one of the most important diseases reducing rice yield. As shown in Fig 6C, when JNX rice was treated with 1% CFB or carbendazim at the heading stage, the neck disease indices were 32.57 ± 2.45% and 33.37 ± 3.45%, respectively. These values were lower compared with the water (69.06 ±3.70%) and the Landy medium (69.03 ± 9.19%) treatments. The thousand-seed weight of the 1% CFB treatment was 32.77 ± 2.45g, similar to that of the carbendazim treatment (32.89 ± 2.45 g); in contrast, the thousand-seed weight of the water treatment was 1.88-fold lower (17.43 ±2.92 g).

## Discussion

At present, the control of rice blast mainly depends on host resistance and application of pesticides. Due to the variation in *M. oryzae* strains, the host resistance is non-sustainable [35, 36].

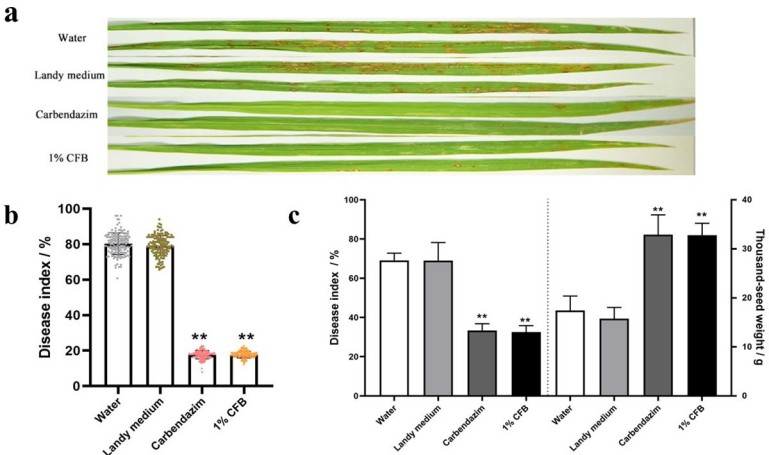

**Fig 6. In the field conditions, the antifungal substances of ZW10 were evaluated regarding control of rice blast.**
The outbreak of leaf blast was different in different treatments (a), resulting in significant treatment differences in disease index (b). The treatment effects on the neck blast disease index and its thousand-seed weight (c). Data are presented as means of three replicates ± SD.

Although chemical control is the dominant strategy, the abuse of pesticides leads to resistance of pathogenic microorganisms and environmental pollution [37]. Therefore, it is an important challenge to seek a sustainable and environmentally friendly control alternatives [38]. *Bacillus* is potentially the ideal strain for biological control because it is environmentally friendly, easy to obtain, and can produce a variety of polypeptide and lipopeptide antifungal metabolites [39–41]. However, *B. velezensis* is a new strain discovered in the recent years, and there are few reports on its biological control of *M. oryzae.*

We isolated and identified a strain of *B. velezensis* from the Sichuan basin neutral purplish soil and named it ZW10; it has antagonistic activity against *M. oryzae* fungi. Based on silica column chromatography, the 60% methanol fraction was the most active, and the IC50 was 1% CFB.

The reason *M. oryzae* can infect plant epidermis is that conidia germinate and produce germ tubes, which grow and expand, and then differentiate into appressorium [2, 42]. After 1% CFB treatment, the conidia germinated and formed germ tubes, but the germination time was relatively prolonged. At 24 h, only 18.98% of conidia treated by 1% CFB germinated to form appressorium, whereas 95.48% of control conidia formed appressorium. These results indicated that *B. velezensis* could delay the germination of conidia and inhibit the formation of appressorium. FDA-PI complex dye was used to detect the activity of *M. oryzae*. It was found that the mycelia treated with 1% CFB showed obvious red fluorescence, especially the deformed mycelia and vacuolar area. In the previous experiment, we found that ZW10 could produce protease, cellulase and chitinase [22]. Therefore, the loss of mycelial permeability caused by 1% CFB treatment of *M. oryzae* may be related to this. Rong et al. [43] reported a similar phenomenon after treatment with the secondary metabolite (Iturin A) of *B. safensis* R2. SEM was used to observe that the mycelium of *M. oryzae* after treatment expanded and that part of the expanded mycelium was broken. At the same time, vacuoles were formed in the middle of the mycelium, and the internal pressure was too high, which led to the rupture of vacuoles and the leakage of cell contents, such as glycerol and trehalose [2]. The blast fungus mechanically breaches the outer plant surface using an appressorium, that generates enormous turgor pressure [3, 44]. And then the appressorium produces a specialized hypha, a penetration peg, which pierces the plant surface [45]. In the experiment with rice sheaths infected by

*M. oryzae*, although 1% CFB treated conidia germinated, they could not form appressoria due to deformations, fractures and formation of vacuoles in germ tubes; hence, they lost the ability to infect rice sheaths. Therefore, we can infer that *B. velezensis* ZW10 affects the permeability of cell membrane of *M. oryzae*, leading to cell disruption and death; in addition, it affects the normal growth of mycelium and spore germination, leading to pathological changes and deformities.

$H_2O_2$ is one of the main biological redox metabolites, whose high concentration induces oxidative damage to biomolecules [46]. Plants can eliminate $H_2O_2$ through enzymatic antioxidants such as peroxidases (POXs), catalases (CATs) and superoxide dismutases (SODs), to avoid oxidative damage to cell structures [47]. In this study, 1% CFB significantly promoted the accumulation of $H_2O_2$ in the LTH leaf cells, which was consistent with the expression of *OsPOD*. $H_2O_2$ and SA are important signaling molecules in the plant defense system; these two molecules can interact with each other. For example, increasing SA can up- regulate the level of hydrogen peroxide in plant tissues [48]. The systemic acquired resistance (SAR) of rice was induced by 1% CFB spraying. The results showed that the expression of *OsPAL1*, *OsLYP6* and *OsPRs* was up-regulated. The transcription factor *WRKY*45 was induced by benzothiadiazole and SA in rice, which helped enhance the host resistance to pathogens [49, 50]. The results showed that the expression trend of *OsWRKY45* was similar to that of the genes involved in the SA pathway.

By producing antibiotics against plant pathogenic microorganisms and activating rice PTI (PAMP-triggered immunity) response, *B. velezensis* directly or indirectly protects against rice blast. The rice variety JNX is susceptible to *M. oryzae*, especially the neck blast. In the field evaluation experiment, leaf blast and neck blast decreased by, respectively, 62.60% and 36.50%, in the 1% CFB treatment, and the thousand-seed weight increased by 188.01%. Hence, using *B. velezensis* is an effective strategy to prevent and control rice blast and reduce the need to applying fungicides.

## Conclusion

In this study, the cell-free culture filtrate of *B. velezensis* ZW10 was purified. The 1% CFB had a significant antagonistic effect against *M. oryzae*. The inhibitory mechanism may involve secretion of active metabolites to directly impair the pathogen or indirectly promote the induction of plant innate immunity. In the field experiment, the antifungal substances of ZW10 significantly reduced the incidence of rice blast, and ultimately increased the yield. In conclusion, ZW10 has a potential to be a biological control agent against rice blast. The future research will focus on the isolation and identification of the active metabolites from ZW10 and the systematic evaluation of them as biological control agents that may be used widely in agriculture. In short, *B. velezensis* ZW10 and its bioactive compounds can be developed as a biopesticide for the biocontrol of rice blast.

## Supporting information

**S1 Fig. Effects of the antifungal substances of ZW10 on *M. oryzae*'s appressorium formation.** After ZW10 treatment, the appressorium was deformed and did not emit green fluoresce.
(TIF)

**S1 File. Germination test of *M. oryzae* conidia and formation of appressorium.** Row date for Fig 2.
(PDF)

**S2 File. Defense-related gene expression.** Row date for Fig 5.
(PDF)

**S3 File. Field trial with the antibacterial antifungal substances of ZW10 against leaf blast and neck blast.** Row date for Fig 6.
(PDF)

**S4 File. 16S rRNA sequence of *B. velezensis* ZW10.**
(PDF)

# Acknowledgments

We are grateful to State Key Laboratory of Crop Gene Exploration and Utilization in Southwest China of Sichuan Agricultural University for providing the rice blast pathogenic fungus *M. oryzae* Guy11 and Guy11-Egfp.

# Author Contributions

**Conceptualization:** Zuo Chen, Lu Zhao, Yilun Dong, Lihua Li, Zhengjun Xu.

**Data curation:** Zuo Chen, Yilun Dong.

**Formal analysis:** Zuo Chen, Lu Zhao, Wenqian Chen, Chunliu Li, Lihua Li.

**Funding acquisition:** Lihua Li, Zhengjun Xu.

**Investigation:** Chunliu Li.

**Methodology:** Zuo Chen, Lu Zhao, Wenqian Chen.

**Project administration:** Chunliu Li.

**Resources:** Zuo Chen, Lu Zhao, Wenqian Chen.

**Software:** Wenqian Chen, Rongjun Chen.

**Supervision:** Rongjun Chen.

**Validation:** Chunliu Li.

**Visualization:** Xiaoling Gao.

**Writing – original draft:** Zuo Chen, Yilun Dong, Rongjun Chen.

**Writing – review & editing:** Xiaoling Gao, Rongjun Chen, Lihua Li, Zhengjun Xu.

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
