## [Decision Letter · Decision Letter 0]

13 Jul 2021

PONE-D-21-17085

The antagonistic mechanism of Bacillus velezensis ZW10 against rice blast disease and its evaluation as a potential biopesticide

PLOS ONE

Dear Dr. Xu,

Thank you for submitting your manuscript to PLOS ONE. After careful consideration, we feel that it has merit but does not fully meet PLOS ONE’s publication criteria as it currently stands. Therefore, we invite you to submit a revised version of the manuscript that addresses the points raised during the review process.

We look forward to receiving your revised manuscript.

Kind regards,

R M Sundaram, Ph.D.

Academic Editor

PLOS ONE

Journal Requirements:

“This work was supported by the Sichuan Science and Technology Program under grant number 2020YJ0352 and grant number 2020YJ0411.”

Additional Editor Comments:

Based on the comments of the reviewers and based on my own review of the manuscript, I recommend it for a major revision

Reviewers' comments:

Reviewer's Responses to Questions

**Comments to the Author**

1. Is the manuscript technically sound, and do the data support the conclusions?

Reviewer #1: Yes

Reviewer #2: Partly

2. Has the statistical analysis been performed appropriately and rigorously? 

Reviewer #1: No

Reviewer #2: Yes

3. Have the authors made all data underlying the findings in their manuscript fully available?

Reviewer #1: Yes

Reviewer #2: Yes

4. Is the manuscript presented in an intelligible fashion and written in standard English?

Reviewer #1: No

Reviewer #2: Yes

5. Review Comments to the Author

Reviewer #1: Comments on MS No. PONE-D-21-17085

MS title: The antagonistic mechanism of Bacillus velezensis ZW10 against rice blast disease and its evaluation as a potential biopesticide

Rice blast caused by Magnaporthe oryzae is one of the most serious diseases of rice causing significant yield loss every year. Extensive work has been done on genetics of blast resistance and several blast resistant rice cultivars are available for commercial cultivation. A large number of highly effective chemicals are also available to manage the disease in the standing crop. However, identification and use of bio-control agents to manage rice blast disease is also equally important.

The authors may note the following points

Line No. 19: it causes the rice grain yield decreases

The author can make: it causes the rice grain yield reduction

Line 25: In the in vivo tests most of M. oryzae could not infect

The sentence should be modified properly

Lines 59-60: In recent years, the research on B. velezensis mainly focused on promoting the growth of animals and plants

The authors may provide reference please

Line 64: Dai Zhang et al.

Should it be only Zhang et al.?

Lines 68-70: Previous studies characterized the physical and chemical properties of B. velezensis ZW10 and elucidated its antagonistic activity against M. oryzae.

Please provide the reference

Line 70: In the paper

Authors can make it ‘in this paper’

Line 76: B. velezensis ZW10 was inoculated to Landy medium (10 L) to be cultured at 35 ℃

The authors are requested to make the sentence proper. The authors should also give the composition of Landy medium or any reference which details the Landy medium composition

Lines 80-84: The authors mentioned that the ethyl acetate extract had the maximum inhibitory effect against the blast fungus and this extract was then further separated using column chromatography and the running phase CH2Cl2 /MeOH at ratio of 40:60 yielded maximum elution of inhibitory compounds. What is the actual compound which gave the inhibitory activity against blast fungus?

Line 91: What was the logic in using GFP-tagged M. oryzae isolate for conidial germination study? Whether the same GFP-tagged M. oryzae isolate was used for field inoculation studies?

Line 96: the germination degree of conidia

The authors can modify the sentence

Lines 107-108: Please give the composition of ‘complete medium’ and any suitable reference. The authors also can mention the objective of this experiment. Is it to know, whether the antagonist produces any cell wall degrading enzymes?

Line 115: Infection of rice leaf sheaths by M. oryzae

The objective of this experiment is not clear

Lines 122-124: Was this experiment was done under glass house condition or inside a plant growth chamber? Whether experiment was done under a particular temperature and RH condition? Whether any treatment was kept with only spraying with CFB (no blast pathogen inoculation)?

Line 172: The fermentation broth was extracted by different polar organic solvents

But hexane is not a polar solvent

Line 189-190: In the treatment group, the germination rate of conidia was only 3.87 ±1.33% after 2 h, and higher than 90% after 8 h.

But Figure 2b shows that germination rate did not cross beyond 20% even after 48 hpi. Please check and clarify. Why in Figure 1b Y-axis it is written formation rate? Should it be germination rate? Overall quality of the figure is not good

Line 216: The antibacterial substances of ZW10 …..

Should it be antifungal?

Figure 5: Pictures/graphs are not clear

Lines 224-228: Upregulation of these genes involved in defense was only due to CFB or pathogen inoculation also has some role? The authors could have taken one treatment with only CFB application.

Lines 235-243: Application of CFB has drastically reduced the leaf and neck blast severity under field condition and equal to carbendazim. Most of the literatures suggest comparatively lower level of protection by biocontrol agents compared to recommended pesticide. How many sprays of CFB were given? The authors could have repeated the field experiment to confirm exceptionally high level of protection by CFB. Graph showing the neck blast control is not clear. Whether any field trial was taken using the antagonist formulation?

Reference # 26 is incompletely cited

Line # 414-415: Ref # 33: 33. Ou SH. Pathogen Variability and Host Resistance in Rice Blast Disease. Ann rev phytopathol. 2014; 18(1):167-187.

Please check the year. It should be 1980

The manuscript requires improvement in its presentation style and language. Also the authors should suggest practical use of this antagonist.

Reviewer #2: Bacillus velezensis is harmless to human being so it should be an excellent candidate as bioagent controlling plant pathogens. The most interesting is that it is widely distributed so will be available easily. But in the present MS there are few issues which are needed for modification. In line no. 44 of introduction you have written which leaf blast and panicle blast are the most common and most harmful: In many places neck blast is emerging in severe form causing massive yield loss. Change the line no 47 as problems in humans and livestock, as well as pathogens are developing resistance to the fungicides instead of resistance to pathogens. Sentence no. 48 & 49 modify the sentence as Therefore, there is a growing interest in the discovery and development of new and improved fungicides based on natural products which are environment friendly as well as the...

Pseudomonas aeruginosa magna has sustained: Please check the species name. besides this is not a good example as Pseudomonas aeruginosa can cause infections in the blood, lungs (pneumonia), or other parts of the body after surgery in human. In line no. 57 B. velezensis, a new species of Bacillus, was discovered in 2005: please give reference. the growth of animals and plants , induction of systemic resistance: Please give reference. 92.07 ± 0.53% after 12 h. In the treatment group, the germination rate of conidia was only 3.87 ± 1.33% after 2 h:In case of blast more than 10,000 spores are formed from a single lesion of a susceptible variety and a susceptible variety contain more than 100 lesions in it’s leaf blades so 1.33% also is not less: How do you justify that? normal appressorium structure due to the abnormal germ tube formation (Fig. 2c):How normal appressorium can be formed due to abnormal germ tube formation? Please rectify the sentence. Please modify all the bibliography in similar style like Annual review of microbiology should be written as Annual Review of Microbiology. Journal of Cereal ence or Journal of Cereal Science?? Frontiers in plant science: please change it to maintain uniform style.

6. PLOS authors have the option to publish the peer review history of their article (what does this mean?). If published, this will include your full peer review and any attached files.

Reviewer #1: No

Reviewer #2: **Yes: **Dr. Arup Kumar Mukherjee

---

## [Author Response · Author response to Decision Letter 0]

22 Jul 2021

Dear Editor and Reviewers:

Thank you for your letter and for the reviewers’ comments concerning our manuscript entitled “The antagonistic mechanism of Bacillus velezensis ZW10 against rice blast disease and its evaluation as a potential biopesticide” (ID: PONE-D-21-17085). Those comments are all valuable and very helpful for revising and improving our paper, as well as the important guiding significance to our researches. We have studied comments carefully and have made correction which we hope meet with approval. Revised portion are marked in red in the paper. The main corrections in the paper and the responds to the reviewer’s comments are as flowing:

Response to the reviewer’s comment:

Reviewer #1：

1. Response to comment: Line No. 19: it causes the rice grain yield decreases.

Response: We have re-written this part according to the reviewer’s suggestion.

2. Response to comment: Line 25: In the in vivo tests most of M. oryzae could not infect. The sentence should be modified properly.

Response: Thanks to the reviewer for pointing out the ambiguous sentences, we have made correction according to the comments.

3. Response to comment: Lines 59-60: In recent years, the research on B. velezensis mainly focused on promoting the growth of animals and plants. The author can make: it causes the rice grain yield reduction. The authors may provide reference please.

Response: Considering the reviewer’s suggestion, we have provided references for evidence.

4. Response to comment: Line 64: Dai Zhang et al. Should it be only Zhang et al.? 

Response: We are very sorry for our incorrect writing. We have corrected it.

5. Response to comment: Lines 68-70: Previous studies characterized the physical and chemical properties of B. velezensis ZW10 and elucidated its antagonistic activity against M. oryzae. Please provide the reference.

Response: Thanks to the reviewer for pointing out the problem, we have provided reference.

6. Response to comment: Line 70: In the paper. Authors can make it ‘in this paper’.

Response: Thank you, we have corrected it. 

7. Response to comment: Line 76: B. velezensis ZW10 was inoculated to Landy medium (10 L) to be cultured at 35 ℃. The authors are requested to make the sentence proper. The authors should also give the composition of Landy medium or any reference which details the Landy medium composition.

Response: Considering the reviewer’s suggestion, we have corrected the sentence and elucidated Landy medium formulation.

8. Response to comment: Lines 80-84: The authors mentioned that the ethyl acetate extract had the maximum inhibitory effect against the blast fungus and this extract was then further separated using column chromatography and the running phase CH2Cl2 /MeOH at ratio of 40:60 yielded maximum elution of inhibitory compounds. What is the actual compound which gave the inhibitory activity against blast fungus?

Response: Thanks to reviewer for point out the problemq. At present, the experiment of separation of B. velezensis ZW10 secondary metabolites is still under planning, so we are sorry that we could not give a definite reply to the reviewer. Refer to relevant papers, the secondary metabolites of Bacillus were as follows surfactin A[1], iturin A[2], this have significant antagonistic effect on pathogenic fungi.

[1]: Jin P, Wang Y, Tan Z, Liu W, Miao W. Antibacterial activity and rice-induced resistance, mediated by C15surfactin A, in controlling rice disease caused by Xanthomonas oryzae pv. oryzae. Pesticide Biochemistry and Physiology. 2020;169:104669.

https://doi.org/10.1016/j.pestbp.2020.104669.

[2]: Rong S, Xu H, Li L, Chen R, Gao X, Xu Z. Antifungal activity of endophytic Bacillus safensis B21 and its potential application as a biopesticide to control rice blast. Pesticide Biochemistry and Physiology. 2020;162:69-77. 

https://doi.org/10.1016/j.pestbp.2019.09.003.

9. Response to comment: Line 91: What was the logic in using GFP-tagged M. oryzae isolate for conidial germination study? Whether the same GFP-tagged M. oryzae isolate was used for field inoculation studies?

Response: Green fluoresce protein (GFP) derived from Aequoreavic-Victoria has the advantages of stable fluorescence property, convenient observation, non-toxic to cells, non-species-specific and no substrate, which can be used for direct and real-time monitoring of the occurrence, colonization and infection process of pathogenic bacteria. It has become one of the important molecular markers in biochemistry and cell biology [1]. Meanwhile, the transfer of nutrients from M. oryzae spore to appressorium could be monitored in real time by fluorescence labeling [2]. And in field inoculation studies, we used common M. oryzae. 

[1] Lippincott-Schwartz J. Development and Use of Fluorescent Protein Markers in Living Cells. Science.2003;300(5616):87-91. https://doi.org/10.1126/science.1082520.

[2] Zhang HF, Zhao Q, Liu KY, Zhang ZG, Wang YC, Zheng XB. MgCRZ1, a transcription factor of Magnaporthe grisea, controls growth, development and is involved in full virulence. Fems Microbiology Letters. 2010; 293(2):160-169. 

https://doi.org/10.1111/j.1574-6968.2009.01524.x.

10. Response to comment: Line 96: the germination degree of conidia. The authors can modify the sentence.

Response: Thank you, we have corrected it.

11. Response to comment: Lines 107-108: Please give the composition of ‘complete medium’ and any suitable reference. The authors also can mention the objective of this experiment. Is it to know, whether the antagonist produces any cell wall degrading enzymes?

Response: Considering the reviewer’s suggestion, we have elucidated Landy medium formulation. In the discussion section, the reason for the result of the experiment is speculated. It is related to the production of protease, cellulase and chitinase by ZW10.

12. Response to comment: Line 115: Infection of rice leaf sheaths by M. oryzae. The objective of this experiment is not clear.

 Response: The blast fungus mechanically breaches the outer plant surface using an appressorium, that generates enormous turgor pressure [1,2]. And then the appressorium produces a specialized hypha, a penetration peg, which pierces the plant surface [3]. Eventually the plant infected with rice blast. Dear rewiewers, we thought that this experiment is helpful readers to understand the process of dynamic infection of GFP-tagged M. oryzae on rice leaf sheath epidermal cells. And the reason why the GFP-tagged M. oryzae cannot infected after 1% CFB treatment. 

[1] Howard R. Breaking and entering: host penetration by the fungal rice blast pathogen magnaporthe grisea. Annual Review of Microbiology. 1996; 50(50):491-512. 

https://doi.org/10.1146/annurev.micro.50.1.491.

[2] Talbot NJ. On the trail of a cereal killer: Exploring the biology of Magnaporthe grisea. Annual Review of Microbiology. 2003;57(1):177-202. 

https://doi.org/10.1146/annurev.micro.57.030502.090957.

[3] Prasanna K, Kirk C, Barbara V. Roles for rice membrane dynamics and plasmodesmata

during biotrophic invasion by the blast fungus. Plant Cell. 2007;19(2):706-724.

https://doi.org/10.1105/tpc.106.046300.

13. Response to comment: Lines 122-124: Was this experiment was done under glass house condition or inside a plant growth chamber? Whether experiment was done under a particular temperature and RH condition? Whether any treatment was kept with only spraying with CFB (no blast pathogen inoculation)?

Response: Thanks to the reviewer for pointing out the deficiencies in writing. The LTH plants were grown in plant growth chamber (18 hours of light at 28 ℃ and 6 hours of darkness at 22 ℃). In this experiment, the control group was sprayed with clean water instead of only spraying with 1% CFB.

14. Response to comment: Line 172: The fermentation broth was extracted by different polar organic solvents. But hexane is not a polar solvent.

Response: We are very sorry for our incorrect writing. We have corrected it.

15. Response to comment: Line 189-190: In the treatment group, the germination rate of conidia was only 3.87 ±1.33% after 2 h, and higher than 90% after 8 h. But Figure 2b shows that germination rate did not cross beyond 20% even after 48 hpi. Please check and clarify. Why in Figure 1b Y-axis it is written formation rate? Should it be germination rate? Overall quality of the figure is not good

Response: Thanks to the reviewer for pointing out the problem. It was caused by an oversight when we processed Fig 2. We have corrected the picture.

16. Response to comment: Line 216: The antibacterial substances of ZW10 … Should it be antifungal?

Response: Thank you, we have made correction according to the comments.

17. Response to comment: Figure 5: Pictures/graphs are not clear.

Response: We are very sorry for our negligence of the clarity Fig 5. We have corrected the picture.

18. Response to comment: Lines 224-228: Upregulation of these genes involved in defense was only due to CFB or pathogen inoculation also has some role? The authors could have taken one treatment with only CFB application.

Response: Plant perception, recognition and response to pathogens are realized by monitoring non self, damaged self and changed self-molecules. Through this mechanism, plants achieve resistance to most pathogens (or potential pathogens) [1]. The experimental design was based on the experimental method of Gao et al. [2]. We hypothesized that these genes might not be activated if they were not inoculated with M. oryzae. 

[1] Sanabria NM, Huang JC, Dubery, IA. Self/non-self perception in plants in innate immunity and defense. Self/Nonself - Immune Recognition and Signaling 2010; 1(1):40-54. https://doi.org/10.4161/self.1.1.10442.

[2] Gao Y, Zeng XD, Ren B, Zeng JR, Xu T, Yang YZ, et al. Antagonistic activity against rice blast disease and elicitation of host-defence response capability of an endophytic Streptomyces albidoflavus OsiLf-2. Plant Pathology. 2020;69(2):259-271. 

https://doi.org/10.1111/ppa.13118. 

19. Response to comment: Lines 235-243: Application of CFB has drastically reduced the leaf and neck blast severity under field condition and equal to carbendazim. Most of the literatures suggest comparatively lower level of protection by biocontrol agents compared to recommended pesticide. How many sprays of CFB were given? The authors could have repeated the field experiment to confirm exceptionally high level of protection by CFB. Graph showing the neck blast control is not clear. Whether any field trial was taken using the antagonist formulation?

Response: In the field experiment, 1% CFB, carbendazim, water, and Landy medium were sprayed 24 hours before spraying M. oryzae. Thanks to the reviewer for pointing out the problem in time. This problem was neglected in writing. The field experiment, the evaluation of leaf blast and panicle and neck blast were all repeated for three times.

20. Response to comment: Reference # 26 is incompletely cited. Line # 414-415: Ref # 33: 33. Ou SH. Pathogen Variability and Host Resistance in Rice Blast Disease. Ann rev phytopathol. 2014; 18(1):167-187. Please check the year. It should be 1980.

Response: Thank you. We have corrected it.

21. Response to comment: The manuscript requires improvement in its presentation style and language. Also the authors should suggest practical use of this antagonist.

 Response: Thanks for the reviewer's correction. We have checked our presentation style and language. We have also modified the conclusion to emphasize the application prospect of this antagonist.

Reviewer #2：

1. Response to comment: In line no. 44 of introduction you have written which leaf blast and panicle blast are the most common and most harmful: In many places neck blast is emerging in severe form causing massive yield loss. 

Response: Thanks for the reviewer's valuable suggestions on revision. We confuse the two proper nouns. we have made correction according to the comments.

2. Response to comment: Change the line no 47 as problems in humans and livestock, as well as pathogens are developing resistance to the fungicides instead of resistance to pathogens. 

Response: Thank you, we have made correction according to the comments.

3. Response to comment: Sentence no. 48 & 49 modify the sentence as Therefore, there is a growing interest in the discovery and development of new and improved fungicides based on natural products which are environment friendly as well as the...

Response: Thanks for the reviewer's valuable suggestions on revision. we have made correction according to the comments.

4. Response to comment: Pseudomonas aeruginosa magna has sustained: Please check the species name. besides this is not a good example as Pseudomonas aeruginosa can cause infections in the blood, lungs (pneumonia), or other parts of the body after surgery in human.

Response: Thanks to the reviewer's correction, we have deleted this part of content.

5. Response to comment: In line no. 57 B. velezensis, a new species of Bacillus, was discovered in 2005: please give reference. The growth of animals and plants, induction of systemic resistance: Please give reference. 

Response: Considering the reviewer’s suggestion, we have provided references for evidence.

 6. Response to comment: In the treatment group, the germination rate of conidia was only 3.87 ± 1.33% after 2 h. In case of blast more than 10,000 spores are formed from a single lesion of a susceptible variety and a susceptible variety contain more than 100 lesions in it’s leaf blades so 1.33% also is not less: How do you justify that?

Response: Thanks for the reviewer's valuable questions. We thought that can be explained by the difference of appressorium activity. As shown in Fig S1, in the control group, appressorium was almost produced after 24 h treatment, and the appressorium also fluoresced. Although few of the ZW10-treated rice blast fungi could produce appressorium, appressorium did not fluorescene, and appressorium (marked) ruptured. Therefore, we hypothesized that appressorium activity decreased after treatment with ZW10. 

Fig S1. Effects of the antifungal substances of ZW10 on M. oryzae’s appressorium formation. After ZW10 treatment, the appressorium was deformed and did not emit green fluoresce. 

7. Response to comment: normal appressorium structure due to the abnormal germ tube formation (Fig. 2c): How normal appressorium can be formed due to abnormal germ tube formation?

 Response: Thanks to the reviewer for pointing out the ambiguous sentences, we have made correction according to the comments.

8. Response to comment: Please rectify the sentence. Please modify all the bibliography in similar style like Annual review of microbiology should be written as Annual Review of Microbiology. Journal of Cereal ence or Journal of Cereal Science? Frontiers in plant science: please change it to maintain uniform style.

Response: Thanks for the reviewer's valuable suggestions on revision. we have made correction according to the comments.

We tried our best to improve the manuscript and made some changes in the manuscript. These changes will not influence the content and framework of the paper. And here we did not list the changes but marked in red in revised paper.

We appreciate for Editor and Reviewers’ warm work earnestly, and hope that the correction will meet with approval.

Once again, thank you very much for your comments and suggestions.

---

## [Decision Letter · Decision Letter 1]

17 Aug 2021

The antagonistic mechanism of Bacillus velezensis ZW10 against rice blast disease：evaluation of ZW10 as a potential biopesticide

PONE-D-21-17085R1

Dear Dr. Xu,

We’re pleased to inform you that your manuscript has been judged scientifically suitable for publication and will be formally accepted for publication once it meets all outstanding technical requirements.

Kind regards,

R M Sundaram, Ph.D.

Academic Editor

PLOS ONE

Additional Editor Comments (optional):

I understand that the authors have addressed the concerns/suggestions of the reviewers. In view of this, I recommend the manuscript for publication in PLoS one

Reviewers' comments:

Reviewer's Responses to Questions

**Comments to the Author**

1. If the authors have adequately addressed your comments raised in a previous round of review and you feel that this manuscript is now acceptable for publication, you may indicate that here to bypass the “Comments to the Author” section, enter your conflict of interest statement in the “Confidential to Editor” section, and submit your "Accept" recommendation.

Reviewer #1: All comments have been addressed

Reviewer #2: All comments have been addressed

2. Is the manuscript technically sound, and do the data support the conclusions?

Reviewer #1: Yes

Reviewer #2: Yes

3. Has the statistical analysis been performed appropriately and rigorously? 

Reviewer #1: Yes

Reviewer #2: Yes

4. Have the authors made all data underlying the findings in their manuscript fully available?

Reviewer #1: Yes

Reviewer #2: Yes

5. Is the manuscript presented in an intelligible fashion and written in standard English?

Reviewer #1: Yes

Reviewer #2: Yes

6. Review Comments to the Author

Reviewer #1: MS # PONE-D-21-17085R1

MS Title: The antagonistic mechanism of Bacillus velezensis ZW10 against rice blast disease：evaluation of ZW10 as a potential biopesticide

Comments

The authors have addressed all the queries and incorporated all the corrections and clarifications raised by the reviewers

Line # 59: was discovered in 2005[13].

Please give a space between 2005 and [13]

Line # 66: antifungal effects of LPs and VOCs released by B. velezensis

Please put the full form of LP and VOC

Line 68: on Xanthomonas oryzae pv. oryzae

pv. will not be in italics

Line # 206: broken, and; moreover

Please make necessary changes

The manuscript may be accepted for publication

Reviewer #2: The article is now modified as per the suggestions. The reference also modified. So, it may be considered for acceptance.

7. PLOS authors have the option to publish the peer review history of their article (what does this mean?). If published, this will include your full peer review and any attached files.

Reviewer #1: No

Reviewer #2: **Yes: **Dr. Arup Kumar Mukherjee

---

## [Editor Report · Acceptance letter]

19 Aug 2021

PONE-D-21-17085R1 

The antagonistic mechanism of *Bacillus velezensis* ZW10 against rice blast disease：evaluation of ZW10 as a potential biopesticide 

Dear Dr. Xu:

I'm pleased to inform you that your manuscript has been deemed suitable for publication in PLOS ONE. Congratulations! Your manuscript is now with our production department. 

Kind regards, 

on behalf of

Dr. R M Sundaram 

Academic Editor

PLOS ONE